# Pathways to psychiatric care in Debre Berhan, Ethiopia: A cross-sectional study

Kaleab Berhanu[1]*, Abayneh Birlie[2], Tizibt Fiseha[1], Yared Reta[3], Yohannes Gebreegziabhere[1]

1 School of Nursing and Midwifery, Asrat Woldeyes Health Sciences College, Debre Berhan University, Debre Berhan, Ethiopia, 2 School of Public Health, Asrat Woldeyes Health Sciences College, Debre Berhan University, Debre Berhan, Ethiopia, 3 College of Medicine and Health Sciences, Hawassa University, Hawassa, Ethiopia

* kalabbrhn@gmail.com

## Abstract

### Background

Pathways to care are the steps individuals went through before finally consulting formal psychiatric services. In developing countries, people with mental disorders (PWMDs) often first consult traditional or religious healers, which may delay treatment. Although studies from different part of Ethiopia confirm this trend, factors influencing indirect pathways remain insufficiently explored.

### Objective

This study aimed to identify pathways to psychiatric care and factors associated with indirect pathways among PWMDs who received psychiatric care from Debre Berhan Comprehensive Specialized Hospital, Ethiopia.

### Methods

We enrolled 446 PWMDs and used the World Health Organization pathway to psychiatric care encounter form to elicit the pathways to psychiatric care. We conducted a multivariable binary logistic regression analysis to identify factors significantly associated with indirect pathways.

### Result

Most of the PWMDs in the study (72.9%) went through indirect pathways. From sociodemographic characteristics, being in the age group between 41–50 years (AOR = 8.27; 95% CI (2.94, 23.18)) and over 50 years (AOR = 6.46; 95% CI (2.00, 20.82)), being female (AOR = 2.51; 95% CI (1.34, 4.73)), being primary school attendees (AOR = 3.00; 95% CI (1.20, 7.40)), being farmer (AOR = 13.00; 95% CI (3.11, 54.31)), and living in the same house with 4–8 people (AOR = 2.77; 95% CI

**Data availability statement:** All relevant data are within the paper and its Supporting Information files.

**Funding:** The author(s) received no specific funding for this work.

**Competing interests:** The authors have declared that no competing interest exist.

**List of abbreviations:** AOR: Adjusted Odds Ratio; CI: Confidence Interval; DALYs: Disability Adjusted Life Year lost; DBCSH: Debre Berhan Comprehensive Specialized Hospital; LMICs: Low-and Middle-Income Countries; MDD: Major Depressive Disorder; PWMDs: People with Mental Disorders; WHO: World Health Organization.

(1.11, 6.95)) were found to be significantly associated with indirect pathways. While from clinical characteristics, a diagnosis of bipolar disorder (AOR = 2.66; 95% CI (1.10, 6.50)) and anxiety (AOR = 3.94; 95% CI (1.37, 11.34)), perceived stigma (AOR = 5.86; 95% CI (3.00, 11.45)), and facing problems during the help-seeking process (AOR = 0.44; 95% CI (0.21, 0.90)) were found to be significantly associated with indirect pathways.

## Conclusion

In this population, PWMDs primarily used indirect pathways as their first point of contact. Several demographic and clinical factors were significantly associated with utilizing indirect pathways. This study has implications for reducing delays by enhancing psychiatric service integration and establishing effective referral systems.

## Introduction

Mental disorders are among the leading causes of global burdens of diseases. Globally, as of 2019, mental and addictive disorders affected more than one billion people worldwide, accounting for 7% of all burdens of diseases as measured in Disability Adjusted Life Years lost (DALYs) and 19% of all Years Lived with Disability [1]. It is estimated that, by the year 2030, mental disorders will account for 25.3% of the DALYs in low-income countries and 33.5% of all the DALYs in middle-income countries [2].

People with mental disorders follow different roots in receiving mental health care. Those events, processes, and intervals before medical treatment are called pathways to care [3]. Pathways usually begin at some identifiable point in the social structure, prompted by the culturally mediated help-seeking interaction between the distressed person and their significant others [4]. Pathways to psychiatric care could be direct or indirect. Direct care is a care provided by a psychiatrist or mental health professionals with no alternative sources. On the contrary, indirect care is a care provided by psychiatrists or mental health professionals after utilizing alternative sources.

In high-income countries, studies showed that a large number of people with mental disorders (PWMDs) receive mental health services directly from mental health professionals in psychiatry clinics or hospitals [5–7]. In contrast, African studies showed that more than half of participants choose traditional and religious healers as their first care providers [8–11].

One of the challenges of utilizing indirect pathways to care is delaying care. In low-income countries, most PWMDs had a longer duration of untreated illness [12,13]. Some of the reasons for this are prioritizing traditional and faith healers as primary sources of help [14], poor mental health infrastructure [15], lacking knowledge about mental health service availability [16], and considering mental illness as trivial [17]. Delay in receiving psychiatric care can lead to an increased level of distress or disability and an increased burden in early detection, identification, and intervention of mental disorders [18,19].

Different studies have attempted to find factors significantly associated with pathways to psychiatric care. A study conducted in Lisbon, Portugal, found that direct pathway was significantly associated with male gender, involuntary admission, referral by a family member, fewer people per room in the household, and lower probability of previous contact with mental health services [6]. In contrast, a study from Italy showed that people with schizophrenia showed a significantly lower rate of self-referral to psychiatric care (40.9%) compared to people with affective (73.57%), neurotic (87.85%), or eating disorders (81.25%) [5]. In England, Preston, a study showed that younger age and suicidal ideation were significantly associated with shorter pathways to direct psychiatric care. On the contrary, being older, being married, having somatic symptoms, and having anxiety and depression diagnoses were associated with longer pathways to direct psychiatry care [20]. Those studies showed that many demographic and clinical characteristics can influence decisions regarding direct or indirect paths to psychiatric care.

Studies conducted in low- and middle-income countries (LMICs) showed that several factors influenced the preference of individual choices of care. A study conducted on Igbo people east of the River Niger, Nigeria, found that higher education predicted preference for the biomedical model. In contrast, low education was associated with traditional and spiritual pathways. In terms of religion, protestants preferred the spiritual path more than Catholics [15]. Another qualitative study conducted in the Delta region of Nigeria found that the reason for choosing indirect pathways was influenced by religious beliefs about treatment (such illness can be cured only by the power of God), traditional beliefs about the causality of mental illness (handiwork of witches, spiritual attacks such as `black magic' and `evil spirits'), poor knowledge of mental health service, and stigma and discrimination [16]. Additionally, a study conducted in Accra, Ghana, found that the odds of first seeking psychiatric care in a non-psychiatric health facility is almost two times higher for the self-employed and four times higher among public servants compared to those who are unemployed [21].

In Ethiopia, studies showed that the community had stigmatizing attitudes towards PWMDs, denied their rights, and prevented them from involving in various responsibilities [22,23]. This might influence pathways one might utilize during a time of crisis. Furthermore, traditional treatment methods were preferred more often for treating symptoms of mental disorders [24]. Additionally, various studies showed low treatment-seeking behavior from formal help sources for mental disorders, while they were preferred more often for physical diseases or symptoms [23–27].

We found four studies about pathways to psychiatric care in Ethiopia [13,24,28–30]. Those studies were conducted in different settings than the one we conducted. Most of the studies were conducted ten years ago, when psychiatric care was not widely available, and the numbers of mental health professionals were scarce. Besides, in the study setting (i.e., Debre Berhan), there are many holy water sites that are considered alternative sources of mental health care. Furthermore, previous studies focused on the factors associated with delay in receiving psychiatric care. However, this study focused on factors associated with indirect pathways to psychiatric care. Hence, this study can add to understanding of pathways to psychiatric care and factors associated with indirect pathways to psychiatric care in Ethiopia.

## Materials and methods

### Study setting, design, and period

This study was conducted after obtaining ethical approval from the ethical review committee of the School of Public Health Asrat Woldeyes Health Sciences campus, Debre Berhan University. The study was conducted at the Debre Berhan Comprehensive Specialized Hospital (DBCSH) psychiatric clinic, located at the center of Debre Berhan town, central Ethiopia. The hospital provides psychiatric care for around five thousand people with different mental disorders each year. We employed an institutional-based cross-sectional study from 30th March to 30th May 2021.

### Study population

We used a single population proportion formula to determine the minimum sample size required for this study. For the proportion of the problem, we used a previous study conducted at Mekelle, Northern Ethiopia (which reported that 74%

of PWMDs utilized indirect pathways) [13]. We also used a 3% margin of error and 95% Confidence Interval (CI). This gave us a minimum initial sample size of 821 PWMDs. However, the total source population for this study is less than 10,000. As a result, we employed a finite population correction formula (i.e., $n = ni/ (1 + ni/N)$). Where "n" is the minimum sample size required for this study, "$n_i$" is the initial minimum sample size calculated (i.e., 821), and "N is the total number of the source population (expected number of PWMDs receiving psychiatric care at DBCSH psychiatric clinic during the study period (i.e., 828 PWMDs) (obtained through personal communication of the head of the department of psychiatry at DBCSH). By adding 10% of the non-response rate, the final minimum sample size required for this study becomes 454 PWMDs.

We employed a consecutive sampling technique to enroll the required minimum sample size. We interviewed all PWMDs, whether new or on follow-up, who were receiving care at the psychiatry department of DBCSH and met the following eligibility criteria: PWMDs who can give consent, able to communicate, and are above 18. Before including participants in the study, we assessed the participant's capacity to consent using a three-item ability to give consent questionnaire [31]. If the participant could provide consent, we got verbal consent and interviewed the patient.

## Measures

### Pathways to psychiatric care

To assess pathways to psychiatric care, we used the WHO pathways to psychiatric care encounter form [32]. The WHO pathway to psychiatric care encounter form is designed to determine pathways to care by documenting the participant's journey. The form has four major pathway contacts, i.e., from where or from whom help is initially sought (the first pathway contact), followed by the second pathway contact, the third pathway contact, and finally, the fourth pathway contact. The WHO encounter form has 22 items to assess pathways that the patients have gone through, including information about the source of referral and duration of the delay to contact.

Fig 1 below shows the structure of the pathways to psychiatric care that the WHO encounter form recommends for developing countries. The form has been used in Ethiopia previously and reported to be acceptable and feasible [13,28,29].

Considering the recommendation of the WHO encounter form for developing countries, we have classified those participants who received psychiatric care directly from modern health care service providers (such as mental health professionals, general practitioners, primary health care providers, and others) for their current mental health disorder as those considered direct path. On the contrary, we classified those participants who received psychiatric care at the DBCSH psychiatric department after visiting alternative sources such as religious healers, traditional healers, herbalists, or other informal sources as those who utilized indirect paths.

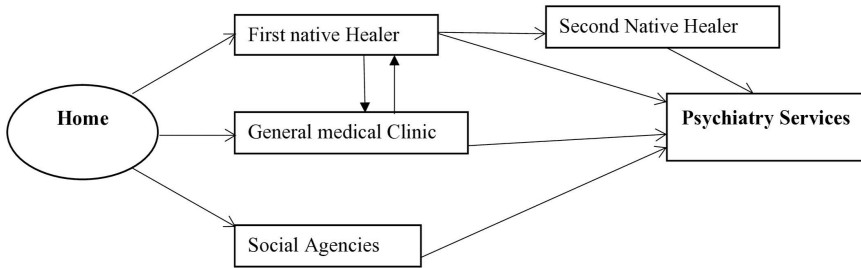

**Fig 1. Schematic presentation of pathways to care in developing countries, adopted from the World Health Organization encounter form.**

## Sociodemographic characteristics

We have used an eleven-item self-developed questionnaire focusing on sociodemographic characteristics such as sex, age, marital status, educational status, and occupational status.

## Social-related factors

We adapted questionnaires used in previous studies to assess social-related factors thought to be significantly associated with pathways to psychiatric care. We evaluated the level of stigma and social support.

For stigma, we used the Jacoby 3-item stigma scale [33]. Jacoby 3-item stigma scale is used to assess perceived stigma among PWMDs, and each item is scored as "0" for "No" and "1" for "Yes" with a total score ranging from 0–3. If a respondent scores one and above, the participant has perceived stigma. The scale was used previously in Ethiopia to assess perceived stigma among PWMDs [34].

For social support, we used the Oslo 3-item social support scale to measure the number and strength of social support [35,36]. The first item of the Oslo 3-item scale scored from 1–4, and the last two scored from 1–5, with a total score ranging from 3–14. A higher score reflects good social support. The Oslo-3 items social support scale has been used in several studies in Ethiopia to measure the number and strength of social support of PWMDs, confirming its feasibility [37,38].

## Clinical-related factors

We developed an eight-item questionnaire to assess clinical-related characteristics not in the WHO pathways to psychiatric care encounter form. We assessed clinical information such as current diagnosis, physical illness, who initiated the current visit, and history of psychiatric consultation.

For delay in psychiatric care, we used the time interval in weeks from onset to contact with psychiatric services. We used the median duration of 52.1 weeks as a reference point to declare treatment delay based on previous studies in Ethiopia [13,29].

For comorbid somatic symptoms commonly reported in PWMDs, we used items from self-report question-20 (SRQ-20) [39]. We include items from SRQ-20 about the presence of somatic symptoms such as headache, abdominal pain, and back pain, including suicidal ideation and attempts. Those somatic symptoms are commonly reported in PWMDs, and studies showed that they are significantly associated with the choice of pathways to psychiatric care [20]. Furthermore, we assessed substance use using self-developed four items about the frequency of use of substances common in the area (i.e., alcohol and khat (an amphitamine like substance)).

## Mental health literacy

We used a ten-item questionnaire to assess the participants' mental health literacy, which the WHO pathways to psychiatric care encounter form did not address. Those items were adapted from previous studies in Ethiopia [13,29]. The items assessed the perception of participants about the cause, treatment, and curability of mental illness.

## Data management and analysis

We checked each questionnaire for completeness and appropriateness of response by the field supervisor each day. We coded and entered each questionnaire into EpiData version 3.1 software, and a double entry was conducted. The entered data were exported to Statistical Package for the Social Sciences (SPSS) version 20 software for quantitative analysis.

We used descriptive statistics (frequency, proportion, mean, and median) to summarize the data and evaluate the distribution of the responses. We checked multicollinearity using the Variance Infiltration Factor (VIF). We explored factors significantly associated with indirect pathways to psychiatric care using binary logistic regression analysis. First, we conducted a bivariable level binary logistic regression analysis and calculated crude odds ratio (COR) with a 95% confidence interval (CI) for all variables included in the bivariable model. Then, we performed a multivariable binary logistic regression

analysis for a variable with a p-value less than 0.25 at the bivariable level analysis to determine the independent effects of each variable. The strength of the association between the dependent and the independent variables was described using an adjusted odds ratio (AOR) with a 95% CI. We summarized the result using texts, tables, and graphs.

## Result

### Sociodemographic characteristics

We included 446 PWMDs with a 98.2% response rate. The reasons for the non-respondents were that three participants interrupted the interview, and five were unwilling to be interviewed. From the total sample, 51.8% (n = 231) were males, with a median age of 33 years (a minimum of 18 and a maximum of 76 years (Table 1).

### Social-related factors

It was observed that 59% (n = 263) of the participants had low social support, 30% (n = 134) had moderate social support, and 11% (n = 49) had strong social support. Additionally, almost half (46%, n = 206) of the participants reported experiencing perceived stigma.

### Clinical related characteristics

Nearly half (49.1%, (n = 219)) of the study participants reported a lifetime history of suicidal ideation, and a little over one-third (31.8% (n = 142)) reported a lifetime history of suicidal attempts. Out of the total participants, 48.4% (n = 215) responded that they have a physical illness; of them, 73.0% (n = 157) have only one physical symptom, and 26.0% (n = 56) have more than one physical illness symptom. The prominent physical illness symptoms reported were weakness or weight loss 26.4% (n = 74), followed by headache (20.3% (n = 57)) (Table 2).

### Mental health literacy-related characteristics

We found that 29.1% (n = 130) of the study participants were delayed in receiving care from modern psychiatric facilities. The top three reasons reported for the delay were not believing in modern treatment (29.6% (n = 88)), not knowing where to seek help (23.6% (n = 70)), and staying in religious healer (holy water) longer (9.1% (n = 27)).

The participants perceived that stress (45.5%, (n = 242)), spiritual possession (18.4%, (n = 98)), and God's will (14.1%, (n = 75)) are the leading causes of mental illness. Almost all (96.0%) of the study participants believed that mental illness is curable. About three-fourths (71.7%, (n = 320)) of the study participants believed modern treatment could treat mental illness.

Regarding the study participants' knowledge about the place where mental health is delivered, more than half of the study participants (59.5%, (n = 266)) responded that they know where the services are provided. About two-thirds (59.6%, (n = 319)) responded that their family knows the symptoms of mental illness (Table 3).

### Pathways to psychiatric care

Only 27.1% (n= (121) of the study participants accessed psychiatric care directly from mental health professionals or other health care providers. The rest, 72.9% with 95% CI: 68.8% – 76.7%, passed through indirect pathways to seek help for their current mental illness (Fig 2).

About one-fifth of the study participants (19.1%, (n = 85)) sought help directly from the psychiatry care service. Most participants (63.4%, (n = 283)) initially chose religious healers as their first point of contact. In their subsequent pathways, 55.8% (n = 241), 22.2% (n = 99), and 2.9% (n = 13) study participants utilized psychiatry care services in their second, third, and fourth pathways, respectively (Fig 2).

In all four pathways, in the majority of the study participants, families/relatives initiated the need to seek care for their current mental illness. In the first pathway, the main reason that led participants to seek help was stress (43%, (n = 193)),

**Table 1. Sociodemographic characteristics of the study participants.**

| Sociodemographic characteristics | Frequency (n = 446) |
|---|---|
| Age in years, % | |
| 18-30 | 42.2 |
| 31-40 | 25.3 |
| 41-50 | 16.8 |
| >50 | 15.7 |
| Sex, % male | 51.8 |
| Marital status, % | |
| Single | 42.8 |
| Married | 41.5 |
| Others* | 15.7 |
| Religion, % Orthodox | 85.9 |
| Ethnicity, % | |
| Amhara | 81.2 |
| Oromo | 15.9 |
| Other* | 2..9 |
| Educational status, % | |
| Unable to read and write | 27.8 |
| Primary school | 24.9 |
| Secondary school | 21.1 |
| College and above | 26.2 |
| Occupational status, % | |
| Civil Servant | 22.0 |
| Unemployed | 39.7 |
| Merchant | 11.0 |
| Farmer | 22.4 |
| Other*** | 4.9 |
| Family size, % | |
| Alone | 10.5 |
| 1-3 | 37.9 |
| 4-8 | 47.1 |
| >8 | 4.5 |
| Residency, % Urban | 72.4 |

*widowed, divorced, and separated bed.

** Tigrie, Gurage, and Wolayta.

*** daily laborer, shoeshine, prisoners.

whereas, in the remaining pathways, the main reason/symptom that led them to seek help was the worsening of their illness (Table 4).

## Factors associated with indirect pathways to psychiatric care

On multivariable binary logistic regression analysis, we found different sociodemographic, social, and clinical characteristics to be significantly associated with utilizing indirect pathways to psychiatric care (Table 5)

From sociodemographic characteristics, age, sex, educational status, occupational status, and family size were significantly associated with utilizing indirect pathways. We found that the odds of those who were in the age group of 41–50

**Table 2. Clinical-related characteristics of the study participants.**

| Variables | Response | Frequency (n = 446) |
|---|---|---|
| Suicidal ideation, ever % yes | | 50.9 |
| Suicidal attempt, ever % yes | | 31.8 |
| Physical illness, % yes | | 48.4 |
| Number of physical illness symptoms | Have one symptom | 73.7 |
| | Have more than one symptom | 26.3 |
| Type of physical illness | Headache, % Yes | 26.5 |
| | weakness and weight loss, % No | 66.0 |
| | Fever, % No | 93.5 |
| | Cough and chest pain, % No | 91.6 |
| | Diabetic mellitus, % No | 88.8 |
| | HIV/AIDS, % No | 94.4 |
| | Hypertension, % No | 91.2 |
| | Gastritis, % No | 84.2 |
| | Other*, % No | 93.5 |
| Visit psychiatry care service before, ever % **yes** | | 33.0 |
| Way of visiting mental health facility, % | Voluntarily | 54.7 |
| | Forced by others | 45.3 |
| Guilty feeling, % yes | | 36.8 |
| Diagnosis, % | Major depressive disorder | 41.7 |
| | Bipolar | 13.7 |
| | Psychosis | 36.8 |
| | Anxiety | 7.8 |
| Lifetime substance use, % yes | | 25.3 |
| Alcohol, % yes | | 70.8 |
| Khat, % yes | | 59.3 |
| Tobacco, % yes | | 43.4 |
| Current substances use, % yes | | 87.6 |
| Alcohol, % | Never | 29.3 |
| | Two to three times a month | 5.1 |
| | Weekly | 14.1 |
| | 3-4 times a week | 28.3 |
| | Daily or Almost Daily (over four times a week) | 23.2 |
| Khat, % | Never | 36.4 |
| | Weekly | 12.1 |
| | 3-4 times a week | 23.2 |
| | Daily or Almost Daily (over four times a week) | 28.3 |
| Tobacco, % | Never | 52.5 |
| | Weekly | 14.1 |
| | 3-4 times a week | 15.2 |
| | Daily or Almost Daily (over four times a week) | 18.2 |

* Abdominal pain, backache, toothache, and skin disease.

** I don't know.

**Table 3. Mental health literacy of the study participants.**

| Variables | Response | Frequency (n = 446) |
|---|---|---|
| Delay in seeking treatment, % > 52 weeks | | 29.1 |
| Reason for delay | Distance of health care facility, % no | 94.8 |
| | Financial difficulties, % no | 95.5 |
| | Did not know where to seek help, % no | 84.3 |
| | Lack of availability of mental health facility, % no | 98.0 |
| | Not believed in modern treatment, % no | 80.3 |
| | Not delayed arrived on early, % no | 59.9 |
| | Because I went to holy water, % no | 94.0 |
| | Other*, % no | 86.5 |
| Problem faced during the help-seeking process, % | Family and friends do not understand the illness | 48.0 |
| | Long waiting times and shortage of medication | 7.6 |
| | Did not face any problem | 44.4 |
| Cause of mental illness | I don't know, % no | 86.1 |
| | God's will, % no | 80.5 |
| | Spiritual possession, % no | 74.5 |
| | Evil eye, % no | 99.0 |
| | Family history, % no | 99.0 |
| | Because I am a sinner, % no | 98.7 |
| | Stress, % no | 37.0 |
| | Magic, % no | 96.2 |
| | Others**, % no | 94.4 |
| Kind of people mental illness affects | Angry and stressed, % yes | 49.3 |
| | People who use drugs, % yes | 43.0 |
| | People with life crisis, % yes | 49.6 |
| | Those who think a lot, % yes | 25.8 |
| | Others***, % no | 94.4 |
| Mental illness is not curable, % no | | 96.0 |
| Know where mental health service is provided, % yes | | 59.6 |
| The appropriate place to receive mental health care service | Modern medicine, % yes | 71.7 |
| | Traditional treatment, % no | 98.0 |
| | Religious treatment, % yes | 55.2 |
| Understanding the severity of mental illness | Very Severe | 65.5 |
| | Less Severe | 28.5 |
| | Not Severe | 6.1 |
| Family members know the symptoms of mental disorder, % yes | | 28.5 |

* I don't know, it will cure by itself, fear of medication side effects and searching for solution by myself

** anger, hormonal change, sorrow, and accident.

*** Anger, being alone.

years were eight times higher in utilizing indirect pathways, while it is six times higher among those who are over 50 years compared to those who are in the age group between 18–30 years (AOR = 8.27; 95% CI (2.94, 23.18) and AOR = 6.46; 95% CI (2.00, 20.82), respectively).

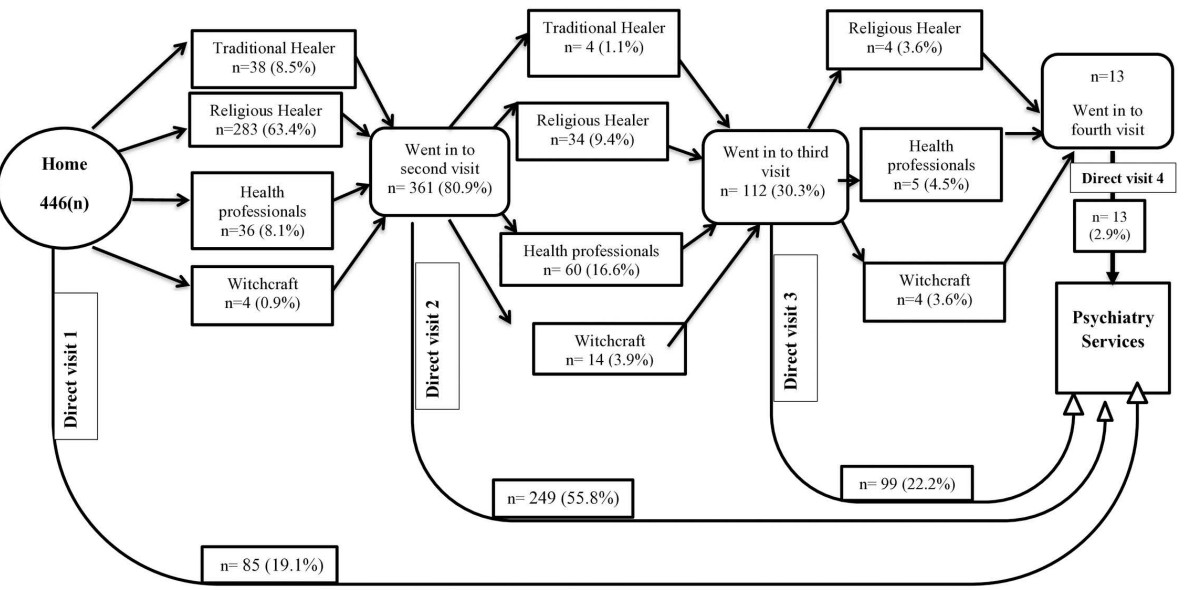

**Fig 2. Diagrammatic presentation of pathways to psychiatric care in Debre Berhan Comprehensive Specialized Hospital Department of Psychiatry, Debre Berhan, Ethiopia, 2021.**

**Table 4. Symptoms and persons that initiated seeking the chosen source of help at different pathways to psychiatric care.**

| Variables | Response | First pathway | Second pathway | Third pathway | Fourth pathway |
|---|---|---|---|---|---|
| | | Frequency (n = 446) (%) | Frequency (n = 361) (%) | Frequency (n = 112) (%) | Frequency (n = 13) (%) |
| Who initiates the path | The patient | 112 (25.1) | 48 (13.3) | 6 (5.4) | 4(30.8) |
| | Family/Relatives | 276 (61.9) | 280 (77.5) | 63 (56.3) | 7(53.8) |
| | Friends | 34 (7.6) | 12 (3.3) | | |
| | Neighbors | 12 (2.7) | 11 (3.0) | 6 (5.3) | |
| | Health professionals | | 8 (2.2) | 37 (33) | 2(15.4) |
| | Other patients | 10 (2.2) | | | |
| | Other* | 2 (0.4) | 2 (0.8) | | |
| Symptoms caused a decision to seek care | Aggressive behavior/harming other | 104 (23.3) | 13 (3.6) | 7 (6.3) | |
| | Suicidal ideation/attempt | 83 (18.6) | 12 (3.3) | 10 (8.9) | |
| | Functional impaired | 17 (3.8) | 9 (2.5) | 6 (5.3) | |
| | Worsening of symptoms | 10 (2.2) | 311 (86.2) | 70 (62.5) | 13 (100) |
| | Stress | 193 (43.3) | | | |
| | Comorbid medical illness | 8 (1.8) | 12 (3.3) | 4 (3.6) | |
| | Unable to sleep | 13 (2.9) | | | |
| | Other** | 18 (4.0) | 4 (1.1) | 15 (13.4) | |

* For Police.

**For mute, talkativeness, and hearing voices.

**Table 5. Bivariate and multivariable binary logistic regression analysis of factors associated with indirect pathways for care.**

| Variables | Responses | Pathways | | COR (CI) | AOR (CI) |
|---|---|---|---|---|---|
| | | Direct | Indirect | | |
| Age in year | 18-30 | 75 | 113 | R | R |
| | 31-40 | 29 | 84 | 1.92 (1.15, 3.21) | 1.62 (0.74, 3.52) |
| | 41-50 | 9 | 66 | **4.87 (2.29, 10.36)** | **8.27 (2.94, 23.18)** |
| | >50 | 8 | 62 | **4.87 (1.98, 11.99)** | **6.46 (2.00, 20.82)** |
| Sex | Male | 75 | 156 | R | R |
| | Female | 46 | 169 | **1.77 (1.15, 2.71)** | **2.51 (1.34, 4.73)** |
| Educational status | Unable to read and write | 15 | 109 | 3.77 (1.95, 7.31) | 0.89 (0.31, 2.49) |
| | Primary school | 21 | 90 | **2.23 (1.21, 4.10)** | **3.00 (1.20, 7.40)** |
| | Secondary school | 45 | 49 | 0.57 (0.32, 0.99) | 0. 50 (0.23, 1.10) |
| | College and above | 40 | 77 | R | R |
| Occupational status | Civil Servant | 35 | 63 | R | R |
| | Unemployed | 60 | 117 | 1.01 (0.65, 1.82) | 0.79 (0.37, 1.67) |
| | Merchant | 13 | 36 | 1.54 (0.72, 3.28) | 1.50 (0.52, 4.31) |
| | Farmer | 8 | 92 | **10.56 (3.92, 28.40)** | **13.00 (3.11, 54.31)** |
| | Other** | 8 | 14 | 0.97 (0.37, 2.54) | 2.28 (0.53, 9.83) |
| Family size | Alone | 23 | 24 | R | R |
| | 1-3 | 46 | 123 | **2.56 (1.39, 4.98)** | **3.21 (1.23, 8.35)** |
| | 4-8 | 50 | 160 | **3.07 (1.59, 5.90)** | **2.77 (1.11, 6.95)** |
| | >8 | 6 | 14 | **2.24 (1.80, 41.41)** | 2.53 (0.27, 23.98) |
| Stigma | Not stigmatized | 87 | 154 | R | R |
| | Stigmatized | 34 | 171 | **2.841 (1.81, 4.47)** | **5.86 (3.00, 11.45)** |
| Diagnosis | MDD | 56 | 130 | R | R |
| | Bipolar | 17 | 44 | 1.11 (0.59, 2.12) | **2.66 (1.10, 6.50)** |
| | Schizophrenia | 36 | 128 | 1.33 (0.76, 2.32) | 1.43 (0.72, 2.84) |
| | Anxiety | 12 | 23 | 0.83(.38, 1.77) | **3.94 (1.37, 11.34)** |
| Problems encountered during help-seeking | Families and friends do not understand the illness | 45 | 169 | R | R |
| | Long waiting times and shortage of medicine | 7 | 27 | 1.03 (0.67, 5.96) | 4.27 (0.92, 19.77) |
| | Didn't face any problem | 72 | 126 | **0.47 (0.30, 0.72)** | **0.44 (0.21, 0.90)** |

* Divorced, widowed, and separated bed.

** daily laborer, shoeshine, prisoners,

*** Police, I do not know.

Bold is for variables with p < 0.05.

P value of Hosmer Lemeshow = 0.062.

In terms of sex, we found that the odds of being female in using indirect pathways were two times higher compared to males (AOR = 2.51; 95% CI (1.34, 4.73)). The educational status of the participants was also found to be significantly associated with utilizing indirect pathways to psychiatric service, where the odds of those who attended primary school were three times higher in utilizing indirect pathways to receive care for their current mental health problem compared with those who attended college and above (AOR = 3.00; 95% CI (1.20, 7.40)).

Occupational status was also significantly associated with utilizing indirect pathways, on which farmers were found to have higher odds of using indirect pathways to psychiatric care compared to civil servants (AOR = 13.00; 95% CI (3.11, 54.31)). The last sociodemographic variable associated with using indirect pathways was family size. We found that the

odds of those who are living with 1–3 people and 4–8 people in the same house to utilize indirect pathways to psychiatric care were three times higher compared to those who were living alone (AOR = 3.21; 95% CI (1.23, 8.35) and (AOR = 2.77; 95% CI (1.11, 6.95), respectively).

Among the social factors, perceived stigma was significantly associated with using indirect pathways. We found that the odds of those who reported perceived stigma in using indirect pathways to psychiatric care were six times higher compared with those who reported no perceived stigma (AOR = 5.86; 95% CI: (3.00, 11.45)).

Among the clinical-related factors, the diagnosis given by clinicians was reported to have a statistically significant association with utilizing indirect pathways to psychiatric care. The odds of those PWMDs diagnosed with bipolar disorder and generalized anxiety disorder were three and four times higher in using indirect pathways compared to those who were diagnosed with Major Depressive Disorder (MDD), respectively (AOR = 2.66; 95% CI (1.10, 6.50), (AOR = 3.94; 95% CI (1.37, 11.34), respectively).

Finally, facing problems during the help-seeking process was found to be significantly associated with utilizing indirect pathways to psychiatric care. The odds of using indirect pathways were 66% less likely among those who did not face problems during the help-seeking process compared to those who reported families and friends do not understand their mental illness as a reason for not seeking help (AOR = 0.44; 95% CI (0.21, 0.90)).

## Discussion

Though different studies have attempted to look for factors significantly associated with delay in psychiatric care in Ethiopia, evidence is scarce about the factors significantly associated with indirect pathways to psychiatric care in Ethiopia. Hence, this study aimed to understand pathways to psychiatric care in PWMDs receiving care in the psychiatric department of DBCSH.

In this study, only about a quarter (27%) of PWMDs received psychiatric care directly from modern psychiatric services. In comparison, a high proportion of PWMDs (73%) utilized indirect pathways to psychiatric care. From all the alternative sources, religious healers were utilized most often as the first point of contact for different kinds of mental illness. We also found that the main reason for utilizing the modern psychiatric service on the first visit was stress. However, as the number of pathways increases, the reason for using modern psychiatric services is the worsening of illness (85.2%, 63.1%, and 100% in the second, third, and fourth pathways, respectively). We found that sociodemographic characteristics (such as being older, being female, being less educated, being a farmer, and living with more than one person in a house), perceived stigma, diagnosis (being diagnosed as a case of bipolar and anxiety disorder), and problems faced during the help-seeking process (reporting family members as a problem on the help-seeking behavior) to be significantly associated with indirect pathways to psychiatric care.

We found that about three-fourths of the participants utilized indirect pathways. Most opted for religious healers, followed by traditional healers and other healthcare providers (such as nurses or general practitioners). Only one-fifth of the study participants directly approached psychiatric services as their primary source of care for their existing mental health issues. This finding is consistent with the finding of the previous study conducted in Mekelle (northern Ethiopia) [13] and Jimma (southwestern Ethiopia) [29], which reported that 74.0% and 65.0% of the study participants utilized indirect pathways as their first point of contact, respectively. The possible explanation for these higher proportions of indirect pathways is probably because most PWMDs prefer holy water for getting a cure for their illness. This was also reported in previous studies in Ethiopia [40].

Our finding is higher compared to studies conducted in Butajira (southern Ethiopia) and Addis Ababa (central Ethiopia), which reported that 59% of the study participants utilized indirect pathways [24,28]. The possible reason for the discrepancy might be, in the Butajira study, one the study used key informants as their source of information while we interviewed PWMDs; two, there are a lot of mental health-related projects in the area, for a longer time, which can promote mental health service utilization. In contrast, the study from Addis Ababa was conducted in the largest city in the country, which

has better access to modern mental health services than our study setting. Moreover, within our study setting, there are several well-known holy water sites where alternative sources of healing are readily available; this may encourage PWMDs to seek alternative sources as their first point of contact.

The current study indicates that PWMDs often opt for an indirect pathway to psychiatric services, paralleling the patterns observed in LMICs, where the prevalence of such indirect routes is reported to be between 40% and 78.0% [9,12,15,16,41–43]. Our findings indicate that PWMDs have a greater likelihood of obtaining psychiatric care through indirect means, which contrasts with the lower rates reported in other studies (12–53%) [17,21,44,45]. This difference may be attributed to the well-established healthcare systems in place, which prioritize specialized care for individuals with mental health issues.

Despite a lot of efforts by the Ethiopian Ministry of Health to integrate mental health into routine service, a significant amount of PWMDs seek help primarily from alternative sources. This might be alarming for the ministry since this may lead to delays in getting help, which can lead to delays in seeking care. The finding has implications for the delay in receiving care; in this study, one-third of the participants had a delay in care. Previous studies showed that delay in psychiatric care is associated with worse clinical outcomes [19].

In this study, old age is significantly associated with choosing indirect pathways. This might be because younger people might be better educated and know where modern care is provided. In comparison, older people may be less willing to receive care from mental health services because they are more likely to hold a belief in self-reliance than relying on others to seek help [46]. This finding aligned with studies conducted in England and India, where younger people chose direct pathways [14,20].

Having a lower educational status was also found to be significantly associated with using indirect pathways. This showed that educational status might have a say in the decision to seek help, and the decision might be biased on the knowledge of the cause and treatment of mental illnesses, which will be directly influenced by educational status. Furthermore, in less educated groups, decisions might be made by families, where, in most cases, they will be at the same level of understanding about the causes and treatment of mental illness. In this study, we witnessed that the majority of the participants have a traditional way of explaining the cause and treatment of mental illness, unlike the biomedical model of explanation (which might require higher educational status to understand and utilize fully) [15]. In addition, knowing where modern service is provided might be more accessible for well-educated participants since they can get information from different reading and listening sources. The association between less education and utilizing indirect pathways agreed with a study conducted in Jaipur, India [14].

In this study, being a farmer was significantly associated with higher odds of utilizing indirect pathways than being a civil servant. This might be because most of the farmers in this study have low educational status (unable to read-write) (67%) and come from rural areas (78%) where religious healers (holy water) were widely available. Similarly, a study conducted in Accra, Ghana, reported that self-employed and public servants have higher odds of utilizing a direct path [21].

We also found that being female holds higher odds of utilizing indirect pathways to psychiatric care. This result was consistent with a study conducted in Lisbon, Portugal, which found that being male was associated with higher odds of utilizing direct psychiatric care [6]. A possible explanation might be that women are expected to be shy when expressing their feelings, while men are often more aggressive. The other reason might be associated with decision-making and knowing where the service is provided.

In addition to the aforementioned sociodemographic characteristics, we also found that larger family size is significantly associated with utilizing indirect pathways to psychiatric care. The possible reason might be when more than one person is involved in the decision to seek care PWMDs might be forced to visit indirect pathways. This result is consistent with a study conducted in Lisbon, Portugal, which found that living with fewer people in the same household was significantly associated with receiving direct psychiatric care [6].

The other factor found to be significantly associated with higher odds of utilizing indirect pathways to psychiatric care was having perceived stigma. This might result from anticipated negative labeling and judgmental reactions from others if one seeks help [47]. This finding was in line with the study conducted in the Delta region of Nigeria [48]. The finding showed that stigma is an additional burden for the health care system by leading participants not to seek help from formal sources directly.

In addition to the factors reported above, the psychiatric diagnosis was also found to be significantly associated with utilizing indirect pathways. We found significantly higher odds of utilizing indirect pathways in PWMDs who are diagnosed with bipolar and generalized anxiety disorders. Partly, it could be because of the presentation of the disorders, where people with bipolar disorder are considered possessed by an evil spirit and need deliverance from spiritual healers. Similar findings were reported in a study conducted in England [44] and Singapore [41].

Finally, we found that those participants who faced problems during the help-seeking process tended to utilize indirect pathways. Stigma, the influence of family members, and lack of access are some of the hindering factors reported in this study, and perhaps this might need to be explored in more detail. We strongly recommend addressing problems in the help-seeking process to improve direct pathways to psychiatric care.

This study is one of the pioneering investigations into the factors influencing the use of indirect pathways to psychiatric care in Ethiopia. Previous research primarily focused on identifying delays in seeking help rather than analyzing the specific patterns of pathways to psychiatric care. Utilizing standardized tools such as the WHO encounter form, Oslo social support, and the 3-item Jacoby scale enabled a comprehensive assessment of various variables that influenced pathways to psychiatric care. However, readers need to acknowledge certain limitations when interpreting the findings. The hospital-based nature of the study may not fully represent the pathways to psychiatric care at the community level for individuals who have not accessed psychiatric care yet. Additionally, potential recall bias could arise due to the nature of mental disorders and the duration of illness among participants. Moreover, data collection through face-to-face interviews by health professionals may introduce social desirability bias.

This study has implications for healthcare providers, caregivers of PWMDs, PWMDs, researchers, and policymakers. For family members, the study has implications for maintaining close relationships with PWMDs within the family in addressing feelings of self-stigma and promoting direct pathways. Future researchers can use this study as a benchmark to design a community-based study to understand the pathways fully. For policymakers, considering policies focused on collaboration between mental health professionals and religious organizations can be beneficial in referring individuals with PWMDs to modern psychiatric services while also incorporating religious therapy. Policymakers and administrators could play their role by enhancing psychiatric services, expanding access at the community level, and integrating modern psychiatric care with religious and traditional healing practices. Hence, they will be able to reduce treatment delays and facilitate easier access to modern mental health services for PWMDs, which in turn improves clinical and functional outcomes.

## Supporting information

**S1. Data Sav.**
(SAV)

## Acknowledgments

We are pleased to extend our heartfelt gratitude to Debre Berhan University Asrat Woldeyes Health Sciences campus for supporting us in conducting this research. Our special thanks go to the Debre Berhan Comprehensive Specialized Hospital Department of Psychiatry staff, who were very supportive and genuine in providing the necessary information and assistance during the data collection. Finally, we would like to express our sincere appreciation to all study participants.

## Author contributions

**Conceptualization:** Kaleab Berhanu, Yohannes Gebreegziabhere.

**Data curation:** Kaleab Berhanu, Yohannes Gebreegziabhere.

**Formal analysis:** Kaleab Berhanu, Tizibit Fisha Mulushewa, Yohannes Gebreegziabhere.

**Funding acquisition:** Kaleab Berhanu.

**Investigation:** Kaleab Berhanu, Abayneh Birlie, Tizibit Fisha Mulushewa, Yared Reta, Yohannes Gebreegziabhere.

**Methodology:** Kaleab Berhanu, Abayneh Birlie, Tizibit Fisha Mulushewa, Yared Reta, Yohannes Gebreegziabhere.

**Project administration:** Kaleab Berhanu, Abayneh Birlie.

**Resources:** Kaleab Berhanu, Abayneh Birlie, Tizibit Fisha Mulushewa, Yared Reta, Yohannes Gebreegziabhere.

**Software:** Kaleab Berhanu.

**Supervision:** Kaleab Berhanu, Tizibit Fisha Mulushewa, Yared Reta, Yohannes Gebreegziabhere.

**Validation:** Kaleab Berhanu, Abayneh Birlie, Tizibit Fisha Mulushewa, Yared Reta, Yohannes Gebreegziabhere.

**Visualization:** Kaleab Berhanu, Abayneh Birlie, Tizibit Fisha Mulushewa, Yared Reta, Yohannes Gebreegziabhere.

**Writing – original draft:** Kaleab Berhanu.

**Writing – review & editing:** Kaleab Berhanu, Abayneh Birlie, Tizibit Fisha Mulushewa, Yared Reta, Yohannes Gebreegziabhere.

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
