## [Decision Letter · Decision Letter 0]

Dear Dr. Kebede,

Thank you for submitting your manuscript to PLOS ONE. After careful consideration, we feel that it has merit but does not fully meet PLOS ONE’s publication criteria as it currently stands. Therefore, we invite you to submit a revised version of the manuscript that addresses the points raised during the review process.

We look forward to receiving your revised manuscript.

Kind regards,

Alemayehu Molla Wollie

Academic Editor

PLOS ONE

Journal Requirements:

2. Please describe in your methods section how capacity to provide consent was determined for the participants in this study. Please also state whether your ethics committee or IRB approved this consent procedure. If you did not assess capacity to consent please briefly outline why this was not necessary in this case.

Additionally, in the ethics statement in the Methods, you have specified that verbal consent was obtained. Please provide additional details regarding how this consent was documented and witnessed, and state whether this was approved by the IRB.

Additional Editor Comments :

Although the manuscript is limited in scope, the authors produced important findings on the research topic.

The manuscript could be improved if authors addressed reviewers’ comments. In addition, please consider the following issues:

Be consistent with terms like informal, indirect, etc.Please include additional sentences that show the gap of the study in the introduction part abstract.Please include additional information like sampling techniques, response rate, etc in the methodology part of the abstract.Currently the focus of mental health treatment is to consider primary care, including traditional healing, to reduce the treatment gap, particularly for low-income countries, but you are recommending the indicated pathway as a negative; please acknowledge such approaches.Please proofread the manuscript for English language and grammar.Please be specific with your study populations (which type of patients) since there are different types of people with symptoms of mental illness.Was the tool validated within a similar population?Please interpret your result in terms of other studies, policies, and guidelines.

Reviewers' comments:

Reviewer's Responses to Questions

**Comments to the Author**

1. Is the manuscript technically sound, and do the data support the conclusions?

Reviewer #1: Yes

Reviewer #2: Partly

2. Has the statistical analysis been performed appropriately and rigorously?

Reviewer #1: Yes

Reviewer #2: Yes

3. Have the authors made all data underlying the findings in their manuscript fully available?

Reviewer #1: Yes

Reviewer #2: Yes

4. Is the manuscript presented in an intelligible fashion and written in standard English?

Reviewer #1: Yes

Reviewer #2: No

Reviewer #1: Dear Authors,

The study you undertake provides valuable insights into the factors influencing pathways to psychiatric care in Debre Berhan, Ethiopia. The authors' comprehensive analysis of sociodemographic characteristics, perceived stigma, diagnosis, and help-seeking behaviors contributes to a better understanding of the challenges faced by individuals with mental health disorders in accessing appropriate care. Below are some concerns you need to go through and react

Introduction part

Concerns

Line 102-103: You mention that little is known about pathways to psychiatric care in Ethiopia, especially indirect pathways. Could you elaborate on the research gap and why this is a significant area to investigate?

Line 111-112: How might the availability of holy water places influence the decision-making process of PWMDs in Ethiopia? Are there any specific beliefs or practices associated with these places that might impact their choice of care?

Line 116-118: How did the WHO pathways to psychiatric care encounter form contribute to the data collection and analysis process? What specific insights did it provide that might not have been possible with other methods?

Method part

Concerns

Line 134: Could you elaborate on the reason for using a 3% margin of error and a 95% confidence interval? Were these values chosen based on specific guidelines or conventions in your field?

Line 138: How was the expected number of PWMDs receiving psychiatric care at DBCSH during the study period obtained? Was this based on previous data, projections, or other sources?

Line 140: Please provide more details about the non-response rate used for the final sample size calculation. How was this rate determined?

Line 142: Can you explain why consecutive sampling was the choice for your research, considering its non-probability characteristics?

Line 145: Could you elaborate on the three-item questionnaire used to assess participant capacity to consent? What were the specific questions asked?

Line 147: Given the potential for bias in verbal consent, consider discussing any measures taken to ensure the validity and reliability of the consent process.

Line 159: Given that the WHO encounter form has been used in previous studies in Ethiopia, what are the specific challenges or limitations that you anticipate encountering in using this tool in your study?

Line 174: Could you provide more details about the specific previous studies that you adapted the questionnaires from for assessing social-related factors? How did you ensure that the adapted questionnaires were appropriate and valid for your study population?

Line 187: Why did you choose to develop your questionnaire to assess clinical-related characteristics, rather than using existing validated instruments? What were the specific factors that you wanted to measure that were not covered by the WHO encounter form?

Line 193: Could you provide more information about the specific items from the SRQ-20 that you included in your study? How did you ensure that these items were relevant and appropriate for assessing comorbid somatic symptoms in your population?

Line 200: What specific mental health conditions were included in the assessment of mental health literacy?

Line 202: Can you provide more details about the WHO pathways to psychiatric care encounter form and how it differs from the ten-item questionnaire?

Line 203: Please elaborate on the specific studies from Ethiopia (references 13 and 29) that were used to adapt the ten-item questionnaire. What were the key similarities and differences between those studies and your research?

Line 204: How did you ensure the ten-item questionnaire adequately assessed participants' perceptions of the cause, treatment, and curability of mental illness? Were there any pilot testing or validation procedures conducted?

Result part

In lines 222-225, you report a high response rate of 98.2% among PWMDs. Could you please provide more details about the characteristics of the non-respondents (e.g., age, gender, disability type) to assess potential biases in your sample? Additionally, could you elaborate on the reasons for the interruptions and unwillingness to participate, as this may shed light on factors influencing participation rates?

Line 285: Could you elaborate on why the adjusted odds ratios for significantly associated variables are substantially higher than their crude counterparts?

Discussion part:

Lines 423-424: Could the authors elaborate on the potential mechanisms through which perceived stigma, family influence, and lack of access contribute to the decision to utilize indirect pathways?

Line 445: Given the prominent role of religious healers in this study, how do the authors envision integrating these practices with modern psychiatric services to improve access to care?

Lines 443-447: Beyond the recommendations for collaboration between mental health professionals and religious organizations, what other policy interventions could be implemented to address the challenges identified in this study?

Better add the followings

1. Study strengths and limitations

2. Recommendation

Reviewer #2: Title

More of the manuscript framed the title to" indirect path psychiatric care". So should modify it.

Abstract:

Methodology: should explain study period and major components are missed. So should rewrite it.

1. What does mean? “We conducted a 25 multivariable binary logistic regression analysis”

2. Key words: add the word “factors”

Introduction

This article aims to provide a comprehensive understanding of indirect path psychiatric care. The introduction statements should be revising to reduce unnecessary sentences, and summarized again. Additionally, grammatical should be corrected to ensure a smooth reading experience. There is no sufficient introduction and explanation about the topic area.

Objective: Not clearly set in the manuscript.

Materials and methods

1. Not clear enough

2. The study setting mental health service types is not clear.

3. Should mention the types of mental health services in the DCSH and explain the average number of psychiatric patients (visits) in every month in the DCSH.

3. Study period Format should be rewrite or stated like Date/month/year.

4. Franckly, sample size calculation is not clear.

5. Both Sampling procedures and sampling techniques are not clear.

6. Eligibility criteria is not clear and Write briefly about the inclusion and exclusion criteria.

7. How many patients were there during the time frame of the study? How many of these patients were not included in the study? Explain clearly why they were not included. If not, it’s difficult to accept the study.

8. Show clearly how it was possible to achieve the actual sample size i .e 454 out of 828 patients in three months.

9. Show clearly how many new cases and follow up cases were during the study period?

Study tools: clear

Source populations: not clear enough Study populations?

Operational definition is not clear

VIF: Write down the value of VIF in number how much it was and also write down the scientific explanation of VIF.

Model of fitness...Not clear, how much it was.

The manuscript has no ethical section. Since its human participant research you should have incorporate the ethical issues in the manuscript.

Result part

1. When the clinical variables were categorized, the authors used the term "psychosis" as a disorder category. This is not a disorder category rather its symptom. So what do you mean “Psychosis" in your study or use the correct disorder category based on ICD-11 or DSM-5 diseases classification manuals.

2. In the associated factors table the P value is not set.

Discussion part

1. Should incorporate the clinical implication of the study finding?

2. The focus of the recommendations should be specific on the study findings.

Example...should recommend about age, being female, and being farmer.

**Do you want your identity to be public for this peer review?** For information about this choice, including consent withdrawal, please see our Privacy Policy

Reviewer #1: **Yes: ** Misrak Negash Shonor

Reviewer #2: No

---

## [Author Response · Author response to Decision Letter 1]

14 Apr 2025

Dear Editors and Reviewers,

We appreciate the time and effort you have put into reviewing our manuscript and providing valuable feedback. We have carefully considered all comments and made the necessary revisions to address them. We have attached our response to each comment and question.

---

## [Decision Letter · Decision Letter 1]

Dear Dr. Kebede,

Thank you for submitting your manuscript to PLOS ONE. After careful consideration, we feel that it has merit but does not fully meet PLOS ONE’s publication criteria as it currently stands. Therefore, we invite you to submit a revised version of the manuscript that addresses the points raised during the review process.

We look forward to receiving your revised manuscript.

Kind regards,

Alemayehu Molla Wollie

Academic Editor

PLOS ONE

Journal Requirements:

Additional Editor Comments:

Dear authors,

I appreciate that you have revised your manuscript accordingly. Since the first reviewers were unavailable, I invited new reviewers to the revised version. They have also considered review reports of original reviewers, and they are satisfied with your revised version despite minor comments below.

Please also clarify ethical issues while interviewing people taking care of their mental illness. I am not sure only oral consent is ethically enough to interview study participants. Once you address this issue and the minor comments of one reviewer, I hope your manuscript will be accepted for publication.

Thanks!

Reviewers' comments:

Reviewer's Responses to Questions

**Comments to the Author**

Reviewer #3: (No Response)

Reviewer #4: All comments have been addressed

2. Is the manuscript technically sound, and do the data support the conclusions?

Reviewer #3: Yes

Reviewer #4: Yes

3. Has the statistical analysis been performed appropriately and rigorously?

Reviewer #3: Yes

Reviewer #4: Yes

4. Have the authors made all data underlying the findings in their manuscript fully available?

Reviewer #3: Yes

Reviewer #4: Yes

5. Is the manuscript presented in an intelligible fashion and written in standard English?

Reviewer #3: Yes

Reviewer #4: Yes

Reviewer #3: Overall: I would like to thank for giving chance to review this interesting work. This study will have great value in low and middle income countries to enhance awareness and prevention strategies for mental health wellbeing. This manuscript reviewed strongly by previous reviewers and authors revised manuscript based on comments. I hope this is good job.

I have few minor comments, find it below

Comments

Abstract section

1. Describe the meaning of pathways to psychiatric care and its impact on mental health

Introduction/Background

1. It is better to explain in detail about indirect Pathways to psychiatric care challenges and its contribution to mental disorders.

2. This study mentioned that ‘‘Debre Berhan) is known for the availability of different holy water places, which are considered an alternative source for mental health care. We expect most PWMDs from various parts of the country to visit those places, hoping for a cure’’. What is relevancy for this statement? Holy water place is easily available in most part of Ethiopia. Why various parts of Ethiopian people going to Debre Berhan for hoping cure? Since holy water is similar in any place. How holy water in Debre Berhan is superior than others place to think most part of country’s people to hope cure. Please remove this statement since holy water is common in all regions in Ethiopia, there is no clear evidence for superiority of holy water in different region.

3. Introductions section is bulky. It is better to make short and precise this section.

Method

1. Participants for this study from patients receiving care in psychiatry clinic during study period. But this study focused indirect contact for psychiatry care, so which contact you assessed current or any contact for help sought? Please describe it clearly.

Result section

1. Other occupational status described as daily laborer, shoeshine, and prisoners. Is it being prisoner categorized as occupation? Modify this.

Finally; - thank you for your contribution to existing body of knowledge’s

Reviewer #4: All my concerns are addressed by previous reviewers, and the authors also addressed them well manner

**Do you want your identity to be public for this peer review?** For information about this choice, including consent withdrawal, please see our Privacy Policy

Reviewer #3: **Yes: ** Tamene Berhanu Alaho

Reviewer #4: No

---

## [Author Response · Author response to Decision Letter 2]

26 Jun 2025

Thank you for your message regarding the metadata (SPSS data file). I apologize for the inconvenience.

I have now re-exported the dataset in .csv. Kindly let me know if there are any further issues or if a different format would be preferred.

Thank you for your comment. As stated in the methods section page 8 line 136-140, we assessed participants' capacity to give consent using a three-item tool to evaluate their ability to give consent before enrollment. Only those who were capable of providing consent were included in the study. Given the minimal risk involved and the nature of the study, we assumed that oral consent was sufficient. The other reason we went for oral consent was that we expected participants to be less educated and have limited ability to read and write, making it challenging to manage written consent procedures (one-third of the study participants are not able to read and write). To ensure consistency and inclusiveness, we chose to obtain oral informed consent after confirming the participant’s capacity to participate in the study. All participants were fully informed about the study’s purpose, their rights, and the voluntary nature of participation. This approach was approved by the Debre Berhan University College of Health Sciences ethical review committee. The meeting and protocol numbers referenced in the ethical committee letter were 1/2021 and 11/12/CHS/SPH.

---

## [Editor Report · Decision Letter 2]

Pathways to Psychiatric Care in Debre Berhan, Ethiopia: a cross-sectional study

PONE-D-24-27256R2

Dear Dr. Kaleab,

We’re pleased to inform you that your manuscript has been judged scientifically suitable for publication and will be formally accepted for publication once it meets all outstanding technical requirements.

Kind regards,

Alemayehu Molla Wollie

Academic Editor

PLOS ONE
---

## [Editor Report · Acceptance letter]

PONE-D-24-27256R2

PLOS ONE

Dear Dr. Kebede,

I'm pleased to inform you that your manuscript has been deemed suitable for publication in PLOS ONE. Congratulations! Your manuscript is now being handed over to our production team.

Kind regards,

on behalf of

Mr. Alemayehu Molla Wollie

Academic Editor

PLOS ONE